# RATT: Recurrent Attention to Transient Tasks for Continual Image Captioning

**Riccardo Del Chiaro**\*
MICC, University of Florence
Florence 50134, FI, Italy
riccardo.delchiaro@unifi.it

**Bartłomiej Twardowski**
CVC, Universitat Autónoma de Barcelona
08193 Barcelona, Spain
bartlomiej.twardowski@cvc.uab.es

**Andrew D. Bagdanov**
MICC, University of Florence
Florence 50134, FI, Italy
andrew.bagdanov@unifi.it

**Joost van de Weijer**
CVC, Universitat Autónoma de Barcelona
08193 Barcelona, Spain
joost@cvc.uab.es

## Abstract

Research on continual learning has led to a variety of approaches to mitigating catastrophic forgetting in feed-forward classification networks. Until now surprisingly little attention has been focused on continual learning of recurrent models applied to problems like image captioning. In this paper we take a systematic look at continual learning of LSTM-based models for image captioning. We propose an attention-based approach that explicitly accommodates the *transient* nature of vocabularies in continual image captioning tasks – i.e. that task vocabularies are not disjoint. We call our method Recurrent Attention to Transient Tasks (RATT), and also show how to adapt continual learning approaches based on weight regularization and knowledge distillation to recurrent continual learning problems. We apply our approaches to incremental image captioning problem on two new continual learning benchmarks we define using the MS-COCO and Flickr30 datasets. Our results demonstrate that RATT is able to sequentially learn five captioning tasks while incurring *no* forgetting of previously learned ones.

## 1 Introduction

Classical supervised learning systems acquire knowledge by providing them with a set of annotated training samples from a task, which for classifiers is a single set of classes to learn. This view of supervised learning stands in stark contrast with how humans acquire knowledge, which is instead *continual* in the sense that mastering new tasks builds upon previous knowledge acquired when learning previous ones. This type of learning is referred to as *continual* learning (sometimes *incremental* or *lifelong* learning), and continual learning systems instead consume a sequence of tasks, each containing its own set of classes to be learned. Through a sequence of *learning sessions*, in which the learner has access only to labeled examples from the current task, the learning system should integrate knowledge from past and current tasks in order to accurately master them all in the end. A principal shortcoming of state-of-the-art learning systems in the continual learning regime is the phenomenon of *catastrophic forgetting* [10, 17]: in the absence of training samples from previous tasks, the learner is likely to *forget* them in the process of acquiring new ones.

Continual learning research has until now concentrated primarily on classification problems modeled with deep, feed-forward neural networks [8, 28]. Given the importance of recurrent networks for

many learning problems, it is surprising that continual learning of recurrent networks has received so little attention [6, 40]. A recent study on catastrophic forgetting in deep LSTM networks [35] observes that forgetting is more pronounced than in feed-forward networks. This is caused by the recurrent connections which amplify each small change in the weights. In this paper, we consider continual learning for captioning, where a recurrent network (LSTM) is used to produce the output sentence describing an image. Rather than having access to all captions jointly during training, we consider different captioning tasks which are learned in a sequential manner (examples of tasks could be captioning of sports, weddings, news, etc).

Most continual learning settings consider tasks that each contain a set of classes, and these sets are disjoint [30, 32, 37]. A key aspect of continual learning for image captioning is the fact that tasks are naturally split into overlapping vocabularies. Task vocabularies might contain nouns and some verbs which are specific to a task, however many of the words (adjectives, adverbs, and articles) are *shared* among tasks. Moreover, the presence of homonyms in different tasks might directly lead to forgetting of previously acquired concepts. This *transient* nature of words in task vocabularies makes continual learning in image captioning networks different from traditional continual learning.

In this paper we take a systematic look at continual learning for image captioning problems using recurrent, LSTM networks. We consider three of the principal classes of approaches to exemplar-free continual learning: weight-regularization approaches, exemplified by Elastic Weight Consolidation (EWC) [17]; knowledge distillation approaches, exemplified by Learning without Forgetting (LwF) [19]; and attention-based approached like Hard Attention to the Task (HAT) [37]. For each we propose modifications specific to their application to recurrent LSTM networks, in general, and more specifically to image captioning in the presence of transient task vocabularies.

The contributions of this work are threefold: (1) we propose a new framework and splitting methodologies for modeling continual learning of sequential generation problems like image captioning; (2) we propose an approach to continual learning in recurrent networks based on transient attention masks that reflect the transient nature of the vocabularies underlying continual image captioning; and (3) we support our conclusions with extensive experimental evaluation on our new continual image captioning benchmarks and compare our proposed approach to continual learning baselines based on weight regularization and knowledge distillation. To the best of our knowledge we are the first to consider continual learning of sequential models in the presence of *transient tasks vocabularies* whose classes may appear in some learning sessions, then disappear, only to reappear in later ones.

## 2   Related work

**Catastrophic forgetting**. Early works demonstrating the inability of networks to retain knowledge from previously task when learning new ones are [27] and [10]. Approaches include methods that mitigate catastrophic forgetting via replay of exemplars (iCarl [32], EEIL [3], and GEM [22]) or by performing pseudo-replay with GAN-generated data [21, 39, 43]. Weight regularization has also been investigated [1, 17, 46]. Output regularization via knowledge distillation was investigated in LwF [19], as well as architectures based on network growing [34, 36] and attention masking [24, 26, 37]. For more details we refer to recent surveys on continual learning [28, 8].

**Image captioning**. Modern captioning techniques are inspired by machine translation and usually employ a CNN image encoder and RNN text decoder to "translate" images into sentences. NIC [42] uses a pre-trained CNN to encode the image and initialize an LSTM decoder. Differently, in [25] image features are used at each time step, while in [9] a two-layer LSTM is employed. Recurrent latent variable is introduced in [5], encoding the visual interpretation of previously-generated words and acting as a long-term visual memory during next words generation. In [44] a spatial attention mechanism is introduced: the model is able to focus on specific regions of the image according to the previously generated words. *ReviewerNet* [45] also selects in advance which part of the image will be attended, so that the decoder is aware of it from the beginning. *Areas of Attention* [29] models the dependencies between image regions and generated words given the RNN state. A visual sentinel is introduced in [23] to determine, at each decoding step, if it is important to attend the visual features. The authors of [2] mixed bottom-up attention (implemented with an object detection network in the encoder) and a top-down attention mechanism in the LSTM decoder that attend to the visual features of the salient image regions selected by the encoder. Recently, transformer-based methods [41] have been applied to image captioning [15, 12, 7], which eliminate the LSTM in the decoder.

The focus of this paper is RNN-based captioning architectures and how they are affected by catastrophic forgetting. For more details on image captioning we refer to recent surveys [14, 18].

**Continual learning of recurrent networks**. A fixed expansion layer technique was proposed to mitigate forgetting in RNNs in [6]. A dedicated network layer that exploits sparse coding of RNN hidden state is used to reduce the overlap of pattern representations. In this method the network grows with each new task. A Net2Net technique was used for expanding the RNN in [40]. The method uses GEM [22] for training on a new task, but has several shortcomings: model weights continue to grow and it must retain previous task data in the memory.

Experiments on four synthetic datasets were conducted in [35] to investigate forgetting in LSTM networks. The authors concluded that the LSTM topology has no influence on forgetting. This observations motivated us to take a close look to continual image captioning where the network architecture is more complex and an LSTM is used as a output decoder.

## 3 Continual LSTMs for transient tasks

We first describe our image captioning model and some details of LSTM networks. Then we describe how to apply classical continual learning approaches to LSTM networks.

### 3.1 Image captioning Model

We use a captioning model similar to Neural Image Captioning (NIC) [42]. It is an encoder-decoder network that "translates" an image into a natural language description. It is trained end-to-end, directly maximizing the probability of correct sequential generation:

$$\hat{\theta} = \arg\max_{\theta} \sum_{(I,s)} \log p(s_N | I, s_1, \ldots, s_{N-1}; \theta). \tag{1}$$

where $s = [s_1, \ldots s_N]$ is the target sentence for image $I$, $\theta$ are the model parameters.

The decoder is an LSTM network in which words $s_1, \ldots, s_{n-1}$ are encoded in the hidden state $h_n$ and a linear classifier is used to predict the next word at time step $n$:

$$
\begin{array}{rclr@{\qquad}rclr}
x_0 &=& V\,\mathrm{CNN}(I) & (2) & h_n &=& \mathrm{LSTM}(x_n, h_{n-1}) & (4) \\
x_n &=& S\,s_n & (3) & p_{n+1} &=& C\,h_n & (5)
\end{array}
$$

where $S$ is a word embedding matrix, $s_n$ is the $n$-th word of the ground-truth sentence for image $I$, $C$ is a linear classifier, and $V$ is the visual projection matrix that projects image features from the CNN encoder into the embedding space at time $n = 0$.

The LSTM network is defined by the following equations (for which we omit the bias terms):

$$
\begin{array}{rclr@{\qquad}rclr}
i_n &=& \sigma(W_{ix}x_n + W_{ih}h_{n-1}) & (6) & & & & \\
o_n &=& \sigma(W_{ox}x_n + W_{oh}h_{n-1}) & (7) & h_n &=& o_n \odot c_n & (10) \\
f_n &=& \sigma(W_{fx}x_n + W_{fh}h_{n-1}) & (8) & c_n &=& f_n \odot c_{n-1} + i_n \odot g_n & (11) \\
g_n &=& \tanh(W_{gx}x_n + W_{gh}h_{n-1}) & (9) & & & &
\end{array}
$$

where $\odot$ is the Hadamard (element-wise) product, $\sigma$ the logistic function, $c$ the LSTM cell state. The $W$ matrices are the trainable LSTM parameters related to input $x$ and hidden state $h$, for each gate $i$, $f$, $o$, $g$. The loss used to train the network is the sum of the negative log likelihood of the correct word at each step:

$$\mathcal{L}(x, s) = -\sum_{n=1}^{N} \log p_n(s_n). \tag{12}$$

**Inference.** During training we perform teacher forcing using $n$-th word of the target sentence as input to predict word $n + 1$. At inference time, since we have no target caption, we use the word predicted by the model at the previous step $\arg\max p_n$ as input to the word embedding matrix $S$.

## 3.2 Continual learning of recurrent models

Normally catastrophic forgetting is highlighted in continual learning benchmarks by defining tasks that are mutually disjoint in the classes they contain (i.e. no class belongs to more than one task). For sequential problems like image captioning, however, this is not so easy: sequential learners must classify *words* at each decoding step, and a large vocabulary of *common* words are needed for any practical captioning task.

**Incremental model.** Our models are trained on sequences of captioning tasks, each having different vocabularies. For this reason any captioning model must be able to enlarge its vocabulary. When a new task arrives we add a new column for each new word in the classifier and word embedding matrices. The recurrent network remains untouched because the embedding projects inputs into the same space. The basic approach to adapt to the new task is to fine-tune the network over the new training set. To manage the different classes (words) of each task we have two possibilities: (1) Use different classifier and word embedding matrices for each task; or (2) Use a common, growing classifier and a common, growing word embedding matrix.

The first option has the advantage that each task can benefit from ad hoc weights for the task, potentially initializing from the previous task for the common words. However, it also increases decoder network size consistently with each new task. The second option has the opposite advantage of keeping the dimension of the network bounded, sharing weights for all common words. Because of the nature of the captioning problem, many words will be shared and duplicating both word embedding matrix and classifier for all the common words seems wasteful. Thus we adopt the second alternative. With this approach, the key trick is to *deactivate* classifier weights for words not present in the current task vocabulary.

We use $\hat{\theta}^t$ to denote optimal weights learned for task $t$ on dataset $D_t$. After training on task $t$, we create a new model for task $t + 1$ with expanded weights for classifier and word embedding matrices. We use weights from $\hat{\theta}^t$ to initialize the shared weights of the new model.

## 3.3 Recurrent continual learning baselines

We describe how to adapt two common continual learning approaches, one based on weight regularization and the other on knowledge distillation. We will use these as baselines in our comparison.

**Weight regularization.** A common method to prevent catastrophic forgetting is to apply regularization to important model weights before proceeding to learn a new task [1, 4, 17, 46]. Such methods can be directly applied to recurrent models with little effort. We choose Elastic Weight Consolidation (EWC) [37] as a regularization-based baseline. The key idea of EWC is to limit change to model parameters vital to previously-learned tasks by applying a quadratic penalty to them depending on their importance. Parameter importance is estimated using a diagonal approximation of the Fisher Information Matrix. The additional loss function we minimize when learning task $t$ is:

$$\mathcal{L}_{\text{EWC}}^t(x, S; \theta^t) = \mathcal{L}(x, S) + \lambda \sum_i \tfrac{1}{2} F_i^{t-1} (\theta_i^t - \hat{\theta}_i^{t-1})^2, \tag{13}$$

where $\hat{\theta}^{t-1}$ are the estimated model parameters for the previous task, $\theta^t$ are the model parameters at the current task $t$, $\mathcal{L}(x, S)$ is the standard loss used for fine-tuning the network on task $t$, $i$ indexes the model parameters shared between tasks $t$ and $t - 1$, $F_i^{t-1}$ is the $i$-th element of a diagonal approximation of the Fisher Information Matrix for model after training on task $t - 1$, and $\lambda$ weights the importance of the previous task. We apply Eq. 13 to all trainable weights. Due to the transient nature of words across tasks, we do not expect weight regularization to be optimal since some words are shared and regularization limits the plasticity needed to adjust to a new task.

**Recurrent Learning without Forgetting**. We also apply a knowledge distillation [13] approach inspired by Learning without Forgetting (LwF) [19] on the LSTM decoder network to prevent catastrophic forgetting. The model after training task $t - 1$ is used as a teacher network when fine-tuning on task $t$. The aim is to let the new network freely learn how to classify new words appearing in task $t$ while keeping stable the predicted probabilities for words from previous tasks.

To do this, at each step $n$ of the decoder network the previous decoder is also fed with the data coming from the new task $t$. Note that the input to the LSTM at each step $n$ is the embedding of the $n-$th word in the target caption, and the same embedding is given as input to both teacher and student

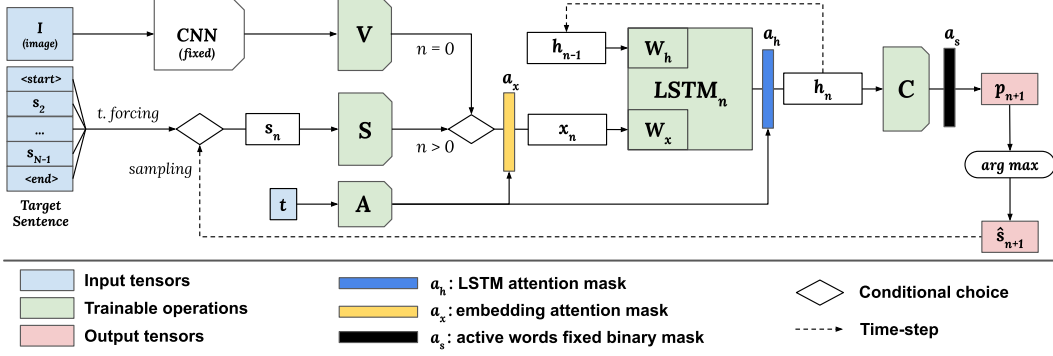

Figure 1: Recurrent Attention to Transient Tasks (RATT). See section 4 for a detailed description of each component of the continual captioning network.

networks – i.e. the student network's embedding of word $n$ is also used as input for the teacher, while each network uses its own hidden state $h_{n-1}$ and cell state $c_{n-1}$ to decode the next word. At each decoding step, the output probabilities $p_{n+1}^{t,<t}$ from the student network LSTM corresponding to words present in tasks $1, \dots, t-1$ are compared with the those predicted by the teacher network, $p_{n+1}^{t-1,<t}$. A distillation loss ensures that the student network does not deviate from the teacher:

$$\mathcal{L}_{\text{LwF}}^t(\hat{p}^t, \hat{p}^{t-1}) \quad = \quad -\sum_n H(\gamma(p_n^{t,<t}), \gamma(p_n^{t-1,<t})) \tag{14}$$

where $\gamma(\cdot)$ rescales a probability vector $p$ -with temperature parameter $T$. This loss is combined with the LSTM training loss (see Eq. 12). Note that differently from [19], we do not fine-tune the classifier of the old network because we use a single, incremental word classifier.

## 4 Attention for continual learning of transient tasks

Inspired by the Hard Attention to the Task (HAT) method [37], we developed an attention-based technique applicable to recurrent networks. We name it *Recurrent Attention to Transient Tasks (RATT)*, since it is specifically designed for recurrent networks with task transience. The key idea is to use an attention mechanism to allocate a portion of the activations of each layer to a *specific* task $t$. An overview of RATT is provided in figure 1.

**Attention masks**. The number of neurons used for a task is limited by two task-conditioned attention masks: embedding attention $a_x^t \in [0, 1]$ and hidden state attention $a_h^t \in [0, 1]$. These are computed with a sigmoid activation $\sigma$ and a positive scaling factor $s$ according to:

$$a_x^t = \sigma(sA_x t^T) , \; a_h^t = \sigma(sA_h t^T), \tag{15}$$

where $t$ is a one-hot task vector, and $A_x$ and $A_h$ are embedding matrices. Next to the two attention mask, we have a vocabulary mask $a_s^t$ which is a binary mask identifying the words of the vocabulary used in task $t$: $a_{s,i}^t = 1$ if word $i$ is part of the vocabulary of task $t$ and is zero otherwise. The forward pass (see Eqs.2 and 5) of the network is modulated with the attention masks according to:

$$\bar{x}_0 = x_0 \odot a_x^t \tag{16} \qquad\qquad \bar{h}_n = h_n \odot a_h^t \tag{18}$$
$$\bar{x}_n = x_n \odot a_x^t \tag{17} \qquad\qquad \bar{p}_{n+1} = p_{n+1} \odot a_s^t \tag{19}$$

Attention masks act as an inhibitor when their value is near 0. The main idea is to learn attention masks during training, and as such learn a limited set of neurons for each task. Neurons used in previous tasks can still be used in subsequent ones, however the weights which were important for previous tasks have reduced plasticity (depending on the amount of attention to for previous tasks).

**Training**. For training we define the cumulative forward mask as:

$$a_x^{<t} = \max(a_x^{t-1}, a_x^{<t-1}), \tag{20}$$

$a_h^{<t}$ and $a_s^{<t}$ are similarly defined. We now define the following backward masks which have the dimensionality of the weight matrices of the network and are used to selectively backpropagate the gradient to the LSTM layers:

$$B_{h,ij}^t = 1 - \min(a_{h,i}^{<t}, a_{h,j}^{<t}) , \; B_{x,ij}^t = 1 - \min(a_{h,i}^{<t}, a_{x,j}^{<t}) \tag{21}$$

Note that we use $a_{h,i}$ refer to the i-th element of vector $a_h$, etc. The backpropagation with learning rate $\lambda$ is then done according to

$$W_h \leftarrow W_h - \lambda B_h^t \odot \frac{\partial \mathcal{L}^t}{\partial W_h} \; , \; W_x \leftarrow W_x - \lambda B_x^t \odot \frac{\partial \mathcal{L}^t}{\partial W_x}. \tag{22}$$

The only difference from standard backpropagation are the backward matrices $B$ which prevents the gradient from changing those weights that were highly attended in previous tasks. The backpropagation updates to the other matrices in Eqs. 6-9 are similar (see Suppl. Materials).

Other than [37] we also define backward masks for the word embedding matrix $S$, the linear classifier $C$, and the image-projection matrix $V$:

$$B_{S,ij}^t = 1 - \min(a_{x,i}^{<t}, a_{s,j}^{<t}) \; , \; B_{C,ij}^t = 1 - \min(a_{s,i}^{<t}, a_{h,j}^{<t}) \; , \; B_{V,ij}^t = 1 - a_{x,i}^{<t}, \tag{23}$$

and the corresponding backpropagation updates:

$$S \leftarrow S - \lambda B_S^t \odot \frac{\partial \mathcal{L}^t}{\partial S} \; , \; C \leftarrow C - \lambda B_C^t \odot \frac{\partial \mathcal{L}^t}{\partial C} \; , \; V \leftarrow V - \lambda B_V^t \odot \frac{\partial \mathcal{L}^t}{\partial V}. \tag{24}$$

The backward mask $B_V^t$ modulates the backpropagation to the image features. Since we do not define a mask on the output of the fixed image encoder, this is only defined by $a_x^{<t}$.

Linearly annealing the scaling parameter $s$, used in Eq. 15, during training (like [37]) was found to be beneficial. We apply $s = \frac{1}{s_{max}} + \left(s_{max} - \frac{1}{s_{max}}\right)\frac{b-1}{B-1}$ where $b$ is the batch index and $B$ is the total number of batches for the epoch. We used $s_{max} = 2000$ and $s_{max} = 400$ for experiments on Flickr30k and MS-COCO, respectively.

The loss used to promote low network usage and to keep some neurons available for future tasks is:

$$\mathcal{L}_a^t = \frac{\sum_i a_{x,i}^t(1 - a_{x,i}^{<t})}{\sum_i(1 - a_{x,i}^{<t})} + \frac{\sum_i a_{h,i}^t(1 - a_{h,i}^{<t})}{\sum_i(1 - a_{h,i}^{<t})}. \tag{25}$$

This loss is combined with Eq. 12 for training. The loss encourages attention to only a few new neurons. However, tasks can attend to previously attended neurons without any penalty. This encourages forward transfer during training. If the attention masks are binary, the system would not suffer from any forgetting, however it would lose its backward transfer ability.

Differently than [37], when computing $B_S^t$ we take into account the recurrency of the network, considering the classifier $C$ to be the previous layer of $S$. In addition, our output masks $a_s$ allow for overlap to model the transient nature of the output vocabularies, whereas [37] only considers non-overlapping classes for the various tasks.

## 5 Experimental results

All experiments use the same architecture: for the encoder network we used ResNet151 [11] pre-trained on ImageNet [33]. Note that the image encoder is frozen and is not trained during continual learning, as is common in many image captioning systems. The decoder consists of the word embedding matrix $S$ that projects the input words into a 256-dimensional space, an LSTM cell with hidden size 512 that takes the word (or image feature for the first step) embeddings as input, and a final fully connected layer $C$ that take as input the hidden state $h_n$ at each LSTM step $n$ and outputs a probability distribution $p_{n+1}$ over the $|V^t|$ words in the vocabulary for current task $t$.

We applied all techniques on the Flickr30K [31] and MS-COCO [20] captioning datasets (see next section for task splits). All experiments were conducted using PyTorch, networks were trained using the Adam [16] optimizer, all hyperparameters were tuned over validation sets. Batch size, learning rate and max-decode length for evaluation were set, respectively, to 128, 4e-4, and 26 for MS-COCO, and 32, 1e-4 and 40 for Flickr30k. These differences are due to the size of the training set and by the average caption lengths in the two datasets.

Inference at test time is *task-aware* for all methods. For EWC and LwF this means that we consider only the word classifier outputs corresponding to the correct task, and for RATT that we use the fixed output masks for the correct task. All metrics where computed using the nlg-eval toolkit [38]. Models where trained for a fixed number of epochs and the best model according to BLEU-4 performance on the validation set were chosen for each task. When proceeding to the next task, the best model from the previous task were used as a starting point.

| Task | Train | Valid | Test | Vocab (words) |
|---|---|---|---|---|
| **transport** | 14,266 | 3,431 | 3,431 | 3,116 |
| **animals** | 9,314 | 2,273 | 2,273 | 2,178 |
| **sports** | 10,077 | 2,384 | 2,384 | 1,967 |
| **food** | 7,814 | 1,890 | 1,890 | 2,235 |
| **interior** | 17,541 | 4,340 | 4,340 | 3,741 |
| **total** | 59,012 | 14,318 | 14,318 | 6,344 |

| Task | Train | Valid | Test | Vocab (words) |
|---|---|---|---|---|
| **scene** | 5,000 | 170 | 170 | 2,714 |
| **animals** | 3,312 | 107 | 113 | 1,631 |
| **vehicles** | 4,084 | 123 | 149 | 2,169 |
| **instruments** | 1,290 | 42 | 42 | 848 |
| **total** | 18,283 | 607 | 636 | 4,123 |

(a) MS-COCO task statistics        (b) Flickr30k task statistics

Table 1: Number of images and words per task for our MS-COCO and Flickr30K splits.

## 5.1 Datasets and task splits

For our experiments we use two different captioning datasets: MS-COCO [31] and Flickr30k [20]. We split MS-COCO into tasks using a *disjoint visual categories* procedure. For this we defined five tasks based on disjoint MS-COCO super-categories containing related classes (*transport*, *animals*, *sports*, *food* and *interior*). For Flickr30K we instead used an *incremental visual categories* procedure. Using the visual entities, phrase types, and splits from [31] we identified four tasks: *scene*, *animals*, *vehicles*, *instruments*. In this approach the first task contains a set of visual concepts that can also be appear in future tasks.

Some statistics on number of images and vocabulary size for each task are given in table 1 for both datasets. See the supplementary material for a detailed breakdown of classes appearing in each task and more details on these dataset splits. MS-COCO does not provide a test set, so we randomly selected half of the validation set images and used them for testing only. Since images have at least five captions, we used the first five captions for each image as the target.

## 5.2 Ablation study

We conducted a preliminary study on our split of MS-COCO to evaluate the impact of our proposed Recurrent Attention to Transient Tasks (RATT) approach. In this experiment we progressively introduce the attention masks described in section 4. We start with the basic captioning model with no forgetting mitigation, and so is equivalent to *fine-tuning*. Then we introduce the mask on hidden state $h_n$ of the LSTM (along with the corresponding backward mask), and then the constant binary mask on the classifier that depends on the words of the current task, then the visual and word embedding masks, and finally the combination of all masks.

In figure 2 we plot the BLEU-4 performance of these configurations for each training epoch and each of the five MS-COCO tasks. Note that for later tasks the performance on early epochs (i.e. *before* encountering the task) is noisy as expected – we are evaluating performance on *future* tasks. These results clearly show that applying the mask to LSTM decreases forgetting in the early epochs when learning a new task. However, performance continues to decrease and in some tasks the result is similar to fine-tuning. Even if the LSTM is forced to remember how to manage hidden states for previous tasks, the other parts of the network suffer from catastrophic forgetting. Adding the classifier mask improves the situation, but the main contribution comes from applying the mask to the embedding. Applying all masks we obtain zero or nearly-zero forgetting. This depends on the $s_{max}$ value used during training: in these experiments we use $s_{max} = 400$, which results in zero forgetting of previous tasks. We also conducted an ablation study on the $s_{max}$ parameter. From the

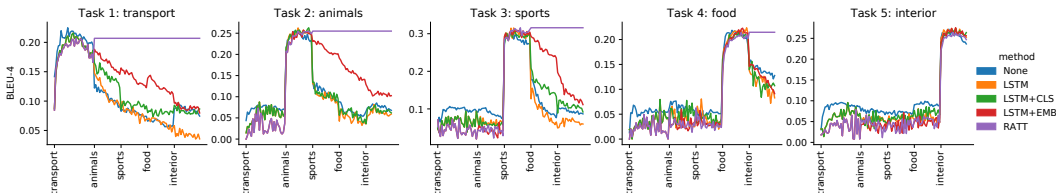

Figure 2: BLEU-4 performance for several ablations at each epoch over the whole sequence of MS-COCO tasks.

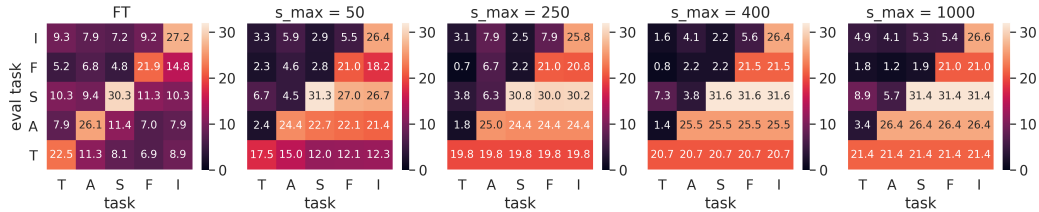

Figure 3: RATT ablation on the MS-COCO validation set using different $s_{max}$ values and fine-tuning baseline. Each heatmap reports BLEU-4 performance for one of the ablated models evaluated on different tasks at the end of the training of each task.

results in figure 3 we can see that higher $s_{max}$ values improve old task performance, and sufficiently high values completely avoid forgetting. Using moderate values, however, can be helpful to increase performance in later tasks. See supplementary material for additional ablations.

## 5.3 Results on MS-COCO

In table 2 we report the performance of a fine-tuning baseline with no forgetting mitigation (FT), EWC, LwF, and RATT on our splits for the MS-COCO captioning dataset. The forgetting percentage is computed by taking the BLEU-4 score for each model after training on the last task and dividing it by the BLEU-4 score at the end of the training of each individual task. From the results we see that all techniques consistently improve performance on previous tasks when compared to the FT baseline. Despite the simplicity of EWC, the improvement over fine-tuning is clear, but it struggles to learn a good model for the last task. LwF instead shows the opposite behavior: it is more capable of learning the last task, but forgetting is more noticeable. RATT achieves *zero* forgetting on MS-COCO, although at the cost of some performance on the final task. This is to be expected, though, as our approach deliberately and progressively limits network capacity to prevent forgetting of old tasks. Qualitative results on MS-COCO are provided in figure 4.

## 5.4 Results on Flickr30k

In table 3 we report performance of a fine-tuning baseline with no forgetting mitigation (FT), EWC, LwF, and RATT on our Flickr30k task splits. Because these splits are based on *incremental visual categories*, it does not reflect a classical continual-learning setup that enforce disjoint categories to maximize catastrophic forgetting: not only there are common words that share the same meaning between different tasks, but some of the visual categories in early tasks are also present in future ones. For this reason, learning how to describe task $t = 1$ also implies learning at least how to partially describe future tasks, so forward and backward transfer is significant.

Despite this, we see that all approaches increase performance on old tasks (when compared to FT) while retaining good performance on the last one. Note that both RATT and LwF result in *negative forgetting*: in these cases the training of a new task results in backward transfer that increases performance on an old one. EWC improvement is marginal, and LwF behaves a bit better and seems more capable of exploiting backward transfer. RATT backward transfer is instead limited by the choice of a high $s_{max}$, which however guarantees nearly zero forgetting.

| | Transport | | | | Animals | | | | Sports | | | | Food | | | | Interior | | | |
|---|---|---|---|---|---|---|---|---|---|---|---|---|---|---|---|---|---|---|---|---|
| | FT | EWC | LwF | RATT | FT | EWC | LwF | RATT | FT | EWC | LwF | RATT | FT | EWC | LwF | RATT | FT | EWC | LwF | RATT |
| **BLEU-4** | .0928 | .1559 | .1277 | **.2126** | .0816 | .1545 | .1050 | **.2468** | .0980 | .2182 | .1491 | **.3161** | .1510 | .1416 | .1623 | **.2169** | .2712 | .2107 | .2537 | **.2727** |
| **METEOR** | .1472 | .1919 | .1708 | **.2169** | .1396 | .1779 | .1577 | **.2349** | .1639 | .2209 | .1918 | **.2707** | .1768 | .1597 | .1962 | **.2110** | **.2351** | .1967 | .2286 | .2257 |
| **CIDEr** | .2067 | .4273 | .3187 | **.6349** | .1480 | .4043 | .2158 | **.7249** | .1680 | .5146 | .3277 | **.8085** | .2668 | .2523 | .3816 | **.5195** | **.6979** | .4878 | .6554 | .6536 |
| **% forgetting** | 59.1 | 31.2 | 43.7 | 0.0 | 67.5 | 33.8 | 45.0 | 0.0 | 68.9 | 23.6 | 45.0 | 0.0 | 32.8 | 14.6 | 16.5 | 0.0 | N/A | N/A | N/A | N/A |

Table 2: Performance on MS-COCO. Numbers are the per-task performance after training on the *last* task. Per-task forgetting in the last row is the BLEU-4 performance after the last task divided by the BLEU-4 performance measured immediately after learning each task.

| | Scene | | | | Animals | | | | Vehicles | | | | Instruments | | | |
|---|---|---|---|---|---|---|---|---|---|---|---|---|---|---|---|---|
| | **FT** | **EWC** | **LwF** | **RATT** | **FT** | **EWC** | **LwF** | **RATT** | **FT** | **EWC** | **LwF** | **RATT** | **FT** | **EWC** | **LwF** | **RATT** |
| **BLEU-4** | .1074 | .1370 | .1504 | **.1548** | .1255 | .1381 | .1384 | **.1921** | .1083 | .1332 | .1450 | **.1724** | .1909 | .2313 | .1862 | **.2386** |
| **METEOR** | .1570 | .1722 | **.1851** | .1710 | .2046 | .1833 | .1954 | **.2107** | .1625 | .1770 | **.1847** | .1750 | **.1933** | .1714 | .1876 | .1782 |
| **CIDEr** | .1222 | .1688 | .2402 | **.2766** | .2460 | .2755 | .2756 | **.4708** | .1586 | .1315 | .1748 | **.2988** | .2525 | .2611 | **.2822** | .2329 |
| **% forgetting** | 31.1 | 11.3 | 2.7 | -2.5 | 38.7 | 19.2 | -15.1 | 0.0 | 35.6 | 4.9 | -1.5 | 0.0 | N/A | N/A | N/A | N/A |

Table 3: Performance on Flickr30K. Evaluation is the same as for MS-COCO.

| | MS-COCO | | | | | Flickr30k | | | |
|---|---|---|---|---|---|---|---|---|---|
| | **T** | **A** | **S** | **F** | **I** | **S** | **A** | **V** | **I** |
| **RATT vs EWC** | 75.0% | 77.5% | 72.5% | 85.0% | 57.5% | 61.8% | 76.4% | 67.3% | 59.5% |
| **RATT vs LwF** | 77.5% | 82.5% | 75.0% | 62.5% | 47.5% | 45.5% | 69.1% | 63.6% | 59.5% |

Table 4: Human captioning evaluation on both MS-COCO and Flickr30k. For each task, we report the percentage of examples for which users preferred the caption generated by RATT.

## 5.5 Human evaluation experiments

We performed an evaluation based on human quality judgments using 200 images (40 from each task) from the MS-COCO test splits. We generated captions with RATT, EWC, and LwF after training on the last task and then presented ten users with an image and RATT and baseline captions in random order. Users were asked (using forced choice) to select which caption best represents the image content. A similar evaluation was performed for the Flickr30k dataset with twelve users. The percentage of users who chose RATT over the baseline are given in the table 4. These results on MS-COCO dataset confirm that RATT is superior on all tasks, while on Flickr30k there is some uncertainty on the first task, especially when comparing RATT with LwF. Note that for the last task of each dataset there is no forgetting, so it is expected that baselines and RATT perform similarly.

## 6 Conclusions

In this paper we proposed a technique for continual learning of image captioning networks based on Recurrent Attention to Transient Tasks (RATT). Our approach is motivated by a feature of image captioning not shared with other continual learning problems: tasks are composed of *transient* classes (words) that can be shared across tasks. We also showed how to adapt Elastic Weight Consolidation and Learning without Forgetting, two representative approaches of continual learning, to the recurrent image captioning networks. We proposed task splits for the MS-COCO and Flickr30k image captioning datasets, and our experimental evaluation confirms the need of recurrent task attention in order to mitigate forgetting in continual learning with sequential, transient tasks. RATT is capable of zero forgetting at the expense of plasticity and backward transfer: the ability to adapt to new tasks is limited by the number of free neurons and it is difficult to exploit knowledge from future tasks to better predict older ones. The focus of this work is on how a simple encoder-decoder image captioning model forget, which limits the quality of captions when comparing with current state-of-the-art. As future work, we are interested in applying the developed method in more complex captioning systems.

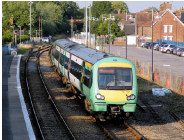

| | |
|---|---|
| **Target** a train riding on a track near a platform | |
| **After training task 1 (Transport):** | **After training task 5 (Interior):** |
| **FT** a train on a track near a platform | **FT** a couple of rows of benches sitting next to a sidewalk |
| **EWC** a train on a track near a platform | **EWC** a train is pulling into a station with people standing on the platform |
| **LwF** a train on a track near a platform | **LwF** a train with a backpack and a backpack on the side of the road |
| **RATT** a train is on the tracks at a station | **RATT** a train is on the tracks at a station |

Figure 4: Qualitative results for image captioning on MS-COCO. Forgetting for baseline methods can be clearly observed. More results are provided in Supplementary Material.

## Broader impact

Automatic image captioning has applications in image indexing, Content-Based Image Retrieval (CBIR), and industries like commerce, education, digital libraries, and web searching. Social media platforms could use it to directly generate descriptions of multimedia content. Image captioning systems can support peoples with disabilities, providing an access to the visual content unreachable before by changing its representation. Continual learning contrasts with the joint-training paradigm in common use today. It has the advantage that it can better protect privacy concerns since, once learned, the data not need to be retained. Furthermore, it is more efficient since networks continue learning and are not initialized from scratch every time a new task arrives. This ability is crucial for development of systems like virtual personal assistants, where the adaption to new tasks and environments is fundamental, especially where multi-modal communication channels are ubiquitous. Finally, the algorithm considered in this paper will reflect the biases present in the dataset. Therefore, special care should be taken when applying this technique to applications where possible biases in the dataset might result in biased outcomes towards minority and/or under-represented groups in the data.

## Acknowledgments and Disclosure of Funding

We acknowledge the support from Huawei Kirin Solution. We thank NVIDIA Corporation for donating the Titan XP GPU that was used to conduct the experiments. We also acknowledge the project PID2019-104174GB-I00 of Ministry of Science of Spain and the ARS01_00421 Project "PON IDEHA - Innovazioni per l'elaborazione dei dati nel settore del Patrimonio Culturale" of the Italian Ministry of Education. Our acknowledged partners funded this project.

## Footnotes

\*Code for experiments available here: https://github.com/delchiaro/RATT

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
