[Supplementary Material]

# RATT: Recurrent Attention to Transient Tasks for Continual Image Captioning
# (SUPPLEMENTARY MATERIAL)

**Riccardo Del Chiaro** *
MICC, University of Florence
Florence 50134, FI, Italy
riccardo.delchiaro@unifi.it

**Bartłomiej Twardowski**
CVC, Universitat Autónoma de Barcelona
08193 Barcelona, Spain
bartlomiej.twardowski@cvc.uab.es

**Andrew D. Bagdanov**
MICC, University of Florence
Florence 50134, FI, Italy
andrew.bagdanov@unifi.it

**Joost van de Weijer**
CVC, Universitat Autónoma de Barcelona
08193 Barcelona, Spain
joost@cvc.uab.es

## 1  Task splits for incremental captioning

Here we first describe the two splitting procedures we propose that are applicable to captioning datasets with categorical annotations. Then we describe how we apply them to the MS-COCO [2] and Flick30k [3] datasets.

### 1.1  Disjoint visual categories

We exploit categorical image annotations available in many captioning datasets. If each image in the dataset belongs to a single category, we can simply define each task as a set of categories that does not overlap with any other task. If an image can belong to multiple categories we instead use the following procedure:

1. **Define $K$ tasks**. Tasks are sets $\mathcal{C}_t$ of categories such that: $\mathcal{C}_i \cap \mathcal{C}_j = \emptyset \; \forall i \neq j$.

2. **Identify candidate example sets**. For each task $t$ select all the examples in the original dataset having at least one of the labels in common with task $t$ categories:

$$\mathrm{P}_t = \{i \mid \exists \, c \in \mathcal{C}_t \text{ s.t. } y_c^i = 1\} \tag{1}$$

   where $i$ is the index of example in the original dataset and $y^i \in \{0, 1\}^{|\mathcal{C}_t|}$ is a multi-label vector such that $y_c^i = 1 \Leftrightarrow$ the $i$-th example belongs to category $c$.

3. **Identify common examples sets**. Find common examples in candidate sets: $\mathrm{Q}_{i,j} = \mathrm{P}_i \cap \mathrm{P}_j$

4. **Define final task examples**. Define example sets of each task $t$ as: $\mathrm{E}_t = \mathrm{P}_t \setminus \cup_{i \neq t}(\mathrm{Q}_{t,i})$

This guarantees that if an image belongs to multiple tasks due to its labels, it will be completely pruned from the dataset instead of added to both or added to only one.

### 1.2  Incremental visual categories

As an alternative to visually-disjoint task splits, we also evaluate continual image captioning in a more real-life setting, where a first task contains a set of visual concepts that can reappear in following

|   | T | A | S | F | I |
|---|---|---|---|---|---|
| **T** | 3,116 (100.0%) | 1,499 (48.11%) | 1,400 (44.93%) | 1,222 (39.22%) | 1,957 (62.80%) |
| **A** | 1,499 (48.11%) | 2,178 (100.0%) | 1,175 (53.95%) | 1,025 (47.06%) | 1,492 (68.50%) |
| **S** | 1,400 (44.93%) | 1,175 (53.95%) | 1,967 (100.0%) | 933 (47.43%) | 1,355 (68.89%) |
| **F** | 1,222 (39.22%) | 1,025 (47.06%) | 933 (47.43%) | 2,235 (100.0%) | 1,530 (68.46%) |
| **I** | 1,957 (62.80%) | 1,492 (68.50%) | 1,355 (68.89%) | 1,530 (68.46%) | 3,741 (100.0%) |

Table 1: word overlaps between tasks for our MS-COCO splits.

tasks. Subsequent tasks contain new or more specific concepts, without the guarantee of having no overlap with the already seen data. The idea is to train the network over general concepts and then progressively train it on more specific ones. The network should continue to perform well on old tasks without overfitting to the more recently seen. The procedure is as follows (note that two first steps are the same as before):

1. **Define $K$ tasks**. Tasks are sets $\mathcal{C}_t$ of categories.
2. **Identify candidate example sets**. As in point (2) of the previous procedure:

$$\mathrm{P}_t = \{i \mid \exists\, c \in \mathcal{C}_t \text{ s.t. } y_c^i = 1\} \tag{2}$$

   where $i$ is the index of an example in the original dataset and $y^i \in \{0,1\}^{|\mathcal{C}_t|}$ is a multi-label vector such that $y_c^i = 1 \Leftrightarrow$ the $i$-th example belongs to category $c$.
3. **Define final task examples**. Define example sets of each task $t$ as: $\mathrm{E}_t = \mathrm{P}_t \setminus \cup_{i=t}^{K}(\mathrm{P}_t \cap \mathrm{P}_i)$

Given the sets $E_t$ we define the training set for the task $t$ as:

$$\mathcal{D}_t = \{x^i, S^{i,1}, S^{i,2}, \dots S^{i,\Sigma} \mid i \in \mathrm{E}_t\} \tag{3}$$

where $S^{i,j}$ is a sentence describing image $x^i$ and $\Sigma$ is the number of sentences describing each image.

## 1.3 An MSCOCO task split

We applied the *disjoint visual categories* splitting procedure to arrive at the following task split for MS-COCO [2]:

- **transport**: bicycle, car, motorcycle, airplane, bus, train, truck, boat.
- **animals**: bird, horse, sheep, cow, elephant, bear, zebra, giraffe.
- **sports**: snowboard, sports ball, kite, baseball bat, baseball glove, skateboard, surfboard, tennis racket.
- **food**: banana, apple, sandwich, orange, broccoli, carrot, hot dog, pizza, donut, cake.
- **interior**: chair, couch, potted plant, bed, toilet, tv, laptop, mouse, remote, keyboard, cell phone, microwave, oven, toaster, sink, refrigerator.

We removed categories *dog* and *cat* from *animals* because objects of these classes are very likely to appear also in images of *interiors* and *sports* tasks. For the same reason we removed *dinning table* from *interior* because of the likely overlap with the *food* task. In table 1 we report word overlaps between tasks for our MS-COCO splits. From this breakdown we see that the task vocabularies are approximately the same size (between around 2,000 and 3,000 words), and there is significant overlap between all tasks.

## 1.4 A Flickr30k task split

In the Flickr30k Entities [3] dataset we have five captions per image and each caption is labeled with a set of *phrase types* that refers to parts of the sentence. We use the union of all phrase types associated to each example as the set of categories for that example. A subset of these categories is used to split the dataset using the *incremental visual categories* procedure. For this dataset we use a single category per task, so tasks are named after assigned categories. The list of categories (tasks) is: **scene**, **animals**, **vehicles**, and **instruments**. If a category is over-represented, random sub-sampling

Figure 1: Flicker30r co-occurrence matrix for assigned categories.

is done to get maximum of 7,500 examples (like in the case of **scene**). Moreover, the most common phrase type is **people** and we omit it in purpose because almost all photos contain people. In figure 1 we give the co-occurrence matrix between Flickr30k images and categories based on phrases types from [3]. The influence of the **people** category is clearly visible.

## 2 Additional training details

In this section give additional details about network training for the approaches we implemented to prevent catastrophic interference in LSTM captioning networks.

### 2.1 Weight Regularization

Optimizing $\mathcal{L}^t_{EWC}(\theta)$ (equation (13) in the main paper) we obtain $\hat{\theta}^t$ and when proceeding to the task $t+1$ we again use $\mathcal{L}^t_{EWC}$ but supplying instead $\hat{\theta}^t_i$ as its argument. $\mathcal{L}^t_{EWC}$ is applied to every trainable weight of our network, but we have special cases for the weights of the word embedding and final classifier: these are only *partially* shared between tasks. At each new task some of the weights will be completely new since they are related to new words, we do not want to force these weights to stay where they are since they have never been trained before, and so we do not regularize them. Note this problem is not present in the standard continual learning for classification because each new task has a disjoint set of classes and a dedicated classifier is used per task.

### 2.2 Recurrent learning without forgetting

The final loss for training the decoder network with LwF is:

$$\mathcal{L}^t(x, S) = \mathcal{L}(x, S) + \mathcal{L}^t_{\text{LwF}}(\hat{p}^t, p^{t-1}) = -\sum_{n=1}^{N} \left[ \log p_n(S_n) - \lambda H(\gamma(\hat{p}^t_n), \gamma(p^{t-1}_n)) \right] \quad (4)$$

where $\lambda$ is the hyperparameter weighting the importance of the previous task. Note that differently from [1], we do not fine-tune the classifier of the old network because we use a single, incremental word classifier.

### 2.3 Recurrent Attention to Trainsent Tasks

The final loss for training the decoder network with RATT is:

$$\mathcal{L}^t(x, S) = \mathcal{L}(x, S) + \mathcal{L}^t_a = -\sum_{n=1}^{N} \log p_n(s_n) + \lambda \frac{\sum_i a^t_{x,i}(1 - a^{<t}_{x,i})}{\sum_i (1 - a^{<t}_{x,i})} + \lambda \frac{\sum_i a^t_{h,i}(1 - a^{<t}_{h,i})}{\sum_i (1 - a^{<t}_{h,i})}. \quad (5)$$

where $\lambda$ is the hyperparameter weighting the importance of future tasks: for larger $\lambda$, fewer neurons will be allocated to the current task (and more neurons will be available for the future tasks).

The backpropagation updates for each LSTM gate matrix are:

$$W_{ih} \quad \leftarrow \quad W_{ih} - \lambda B_{ih}^t \odot \frac{\partial \mathcal{L}^t}{\partial W_{ih}} \quad (6) \qquad W_{fh} \quad \leftarrow \quad W_{fh} - \lambda B_{fh}^t \odot \frac{\partial \mathcal{L}^t}{\partial W_{fh}} \quad (10)$$

$$W_{ix} \quad \leftarrow \quad W_{ix} - \lambda B_{ix}^t \odot \frac{\partial \mathcal{L}^t}{\partial W_{ix}} \quad (7) \qquad W_{fx} \quad \leftarrow \quad W_{fx} - \lambda B_{fx}^t \odot \frac{\partial \mathcal{L}^t}{\partial W_{fx}} \quad (11)$$

$$W_{oh} \quad \leftarrow \quad W_{oh} - \lambda B_{oh}^t \odot \frac{\partial \mathcal{L}^t}{\partial W_{oh}} \quad (8) \qquad W_{gh} \quad \leftarrow \quad W_{gh} - \lambda B_{gh}^t \odot \frac{\partial \mathcal{L}^t}{\partial W_{gh}} \quad (12)$$

$$W_{ox} \quad \leftarrow \quad W_{ox} - \lambda B_{ox}^t \odot \frac{\partial \mathcal{L}^t}{\partial W_{ox}} \quad (9) \qquad W_{gx} \quad \leftarrow \quad W_{gx} - \lambda B_{gx}^t \odot \frac{\partial \mathcal{L}^t}{\partial W_{gx}} \quad (13)$$

During training, we applied the gradient compensation procedure described in [4] to help training the task-embedding matrices $A_x$ and $A_h$:

$$A_{x,i} \quad \leftarrow \quad \frac{s_{max}[\cosh(sA_{x,i}t^T)+1]}{s[\cosh(A_{x,i}t^T)+1]} \frac{\partial \mathcal{L}^t}{\partial A_{x,i}} \quad (14)$$

$$A_{h,i} \quad \leftarrow \quad \frac{s_{max}[\cosh(sA_{h,i}t^T)+1]}{s[\cosh(A_{h,i}t^T)+1]} \frac{\partial \mathcal{L}^t}{\partial A_{h,i}}. \quad (15)$$

Moreover, for numerical stability, we clamp $|s\,A_{x,i}t^T| \leq 50$ and $|s\,A_{h,i}t^T| \leq 50$.

## 3 Additional ablations

In figure 2 we provide a different visualization of the RATT ablation reported in the main paper where we apply attention masking in different layers of the decoder architecture. In figure 2 we observe an increase of performance on old tasks when the classifier mask is used, and even more clearly when the embedding mask is used. Even further improvement in the performance is made when all the attention masks (the RATT approach) are used and there is no forgetting.

We also conducted an ablation study on the $s_{max}$ parameter on Flickr30k, and results are reported in figure 3. Different visualizations for this ablation are shown in figure 4 (for MS-COCO) and 5 (for Flickr30k). From the MS-COCO experiment backward transfer for RATT is not noticeable, while for the Flickr30k case we observe in figure 5 that lower $s_{max}$ values result in a small boost in performance for previous tasks when the training is started on each new one. However at the end of each training session the forgetting is always greater than the backwards transfer. Moreover, the model with highest $s_{max}$ (purple line in figure 5) still shows a small amount of backward transfer, and in this case the performance gain is retained until the end of training. This is also noticeable in the last heatmap of figure 3 for the first task (Sport) (bottom row).

Figure 2: RATT ablation on the MS-COCO validation set using different attention masks. Each heatmap report BLEU-4 performance for one of the ablated models evaluated on different tasks at the end of the training of each task.

Figure 3: RATT ablation on Flickr30k validation set using different $s_{max}$ values and finetuning baseline. Each heatmap reports BLEU-4 performance for one of the ablated models evaluated on different tasks at the end of the training of each task.

Figure 4: RATT ablation on the MS-COCO validation set using different $s_{max}$ values and finetuning baseline. Each plot reports BLEU-4 performance evaluated on one of the tasks at different training epochs and different training tasks for each of the ablated models.

Figure 5: RATT ablation on Flickr30k validation set using different $s_{max}$ values and finetuning baseline. Evaluation is the same as MS-COCO (figure 4).

# 4 Additional experimental analysis

In this section we give additional comparative performance analysis for RATT, EWC, and LwF on both datasets.

## 4.1 Learning and forgetting on MS-COCO

In figures 6 and 7, we give a comparison of performance for all considered approaches on the MS-COCO validation set. These learning curves and heatmaps allow us to appreciate the ability of RATT to remember old tasks. The forgetting rate of EWC seems to be higher than the one of LwF, but EWC shows an ability to recover performance after noticeable forgetting – probably due to increased backward transfer. This is clear looking at figure 7 in which both LwF and EWC seems to suffer noticeable forgetting on the first two tasks (transport and animal) after training on the third one (Sport). EWC seems able to recover when trained on the next task, while LwF continues to forget more.

Figure 6: Comparison for all approaches on MS-COCO validation set. Each plot reports BLEU-4 performance evaluated on one of the tasks at different training epochs and different training tasks for each of the ablated models.

Figure 7: Comparison for all approaches on MS-COCO validation set. Each heatmap reports BLEU-4 performance for one of the models evaluated on different tasks at the end of the training of each task.

Figure 8: Comparison for all approaches on Flickr30k validations set. Each plot reports BLEU-4 performance evaluated on one of the tasks at different training epochs and different training tasks for each of the ablated models.

Figure 9: Comparison for all approaches on Flickr30k validations set. Each heatmap reports BLEU-4 performance for one of the models evaluated on different tasks at the end of the training of each task.

## 4.2 Learning and forgetting on Flickr30k

In figure 8 and 9 we give a comparison of performance for all approaches on the Flickr30k validation set. The first figure depicts the training process over all tasks, where the model is evaluated on each task while progressing through training. The results for Flickr30k show more variance than MS-COCO, as this setting is more challenging and the validation dataset is much smaller.

RATT exhibits almost no forgetting in comparison to other methods – an almost straight line after learning each task. Degradation of the FT model is visible, but for Flickr30k we notice that

subsequent, more specific tasks keep previously learned and more generic concepts rather than completely forgetting (i.e. the first task category **scene**). The BLEU-4 score for LwF remains almost at the same level after learning the task, and EWC shows similar performance but with a bigger drop when going from task A to V. In figure 9 an evaluation summary is provided in form of BLEU-4 heatmaps. Going from the left (FT) to right (RATT), less forgetting can be observed by each of evaluated method, with RATT showing almost no loss in performance when reaching the final task.

It is useful to compare and contrast results on Flickr30k and MS-COCO. In Flickr30k there is much more information shared between tasks and this is shown by the significant forward transfer that we see: after training on the first task (scene), the performance on the last task (instrument) is significant for all methods. Forward transfer is much less evident for RATT, and this is due to the fact that it use the task embedding of future tasks for which it has no information (they all are randomly initialized). The backward transfer on Flickr30k is also evident looking at the relatively high performance of the FT baseline in figures 8 and 9 (and comparing with the MS-COCO equivalents in figures 6 and 7).

Although the overall performance on Flickr30k is much lower than on MS-COCO (evident when looking at the anti-diagonal of FT in figures 7 and 9), given the difficulty of the dataset itself and given the small number of examples (especially in validation/test sets) is difficult to draw firm conclusions about backward transfer for LwF and EWC.

## 5   Additional captioning results

In figure 10 we give an example image from each of the first four MS-COCO tasks with the prediction made by the models after training on the correct task (on the left) and the one made after training on the complete sequence of tasks (on the right). Both EwC and LwF retain some correct words and "insight", but they are clearly confused by the last task on which they are trained. In the second image EWC predicts zebras in a living room because the last task contain house interiors. In a similar way, in the last picture EWC predicts the words *refirgerator* and *bed*, while LwF predicts *table*. In figure 11 we can see a similar analysis conducted on Flickr30k dataset. Again the quality of RATT captions is retained after training on the last task. In figure 12 we give two qualitative examples taken from the last task from the MS-COCO dataset for which fine-tuning provides better descriptions than RATT. In this case the baseline does not suffer from catastrophic forgetting because we evaluate the last trained task. RATT could be limited by the fact that neurons allocated to previous tasks are not trainable.

| **Target** a passenger bus that is driving down the street | |
|---|---|
| **After training task 1 (Transport):** | **After training task 5 (Interior):** |
| **FT**    a bus is stopped at a bus stop | **FT**    a street scene with focus on the wall |
| **EWC**  a bus is stopped at a bus stop | **EWC**  a double decker bus is on the street |
| **LwF**   a bus is stopped at a bus stop | **LwF**   a group of people standing next to each other |
| **RATT** a bus is parked in front of a building | **RATT** a bus is parked in front of a building |

| **Target** a number of zebras standing in the dirt near a wall | |
|---|---|
| **After training task 2 (Animal):** | **After training task 5 (Interior):** |
| **FT**    a group of zebras are standing in a field | **FT**    a woman in a black shirt is walking by a beach |
| **EWC**  a group of zebras standing in a dirt field | **EWC**  a group of zebras are standing in a living room |
| **LwF**   a group of zebras are standing in a field | **LwF**   a black and white photo of a group of people |
| **RATT** a group of zebras are standing in the dirt | **RATT** a group of zebras are standing in the dirt |

| **Target** a man is holding a surfboard and staring out into the ocean | |
|---|---|
| **After training task 3 (Sport):** | **After training task 5 (Interior):** |
| **FT**    a man carrying a surfboard on top of a beach | **FT**    a woman in a black shirt is walking by a beach |
| **EWC**  a man carrying a surfboard on top of a beach | **EWC**  a man riding a surfboard on a beach |
| **LwF**   a man carrying a surfboard on top of a beach | **LwF**   a woman walking down a beach with a umbrella |
| **RATT** a man holding a surfboard in the ocean | **RATT** a man holding a surfboard in the ocean |

| **Target** a woman sells cupcakes with fancy decorations on them | |
|---|---|
| **After training task 4 (Food):** | **After training task 5 (Interior):** |
| **FT**    a woman is standing in front of a table full of food | **FT**    a woman is holding a glass of wine at a restaurant |
| **EWC** a man is holding a banana in a kitchen | **EWC** a woman is holding a white refrigerator in a bed |
| **LwF**  a woman standing in front of a store with a large crowd of people | **LwF**  a woman standing in front of a table with a large pot of food |
| **RATT** a woman standing in front of a store filled with cakes | **RATT** a woman standing in front of a store filled with cakes |

Figure 10:  Captioning results for all methods on MS-COCO. Images and target captions belong to a specific task and captions are generated by all techniques after training the correct task (left) and a later task (right). Approaches except RATT contextualize to some degree generated captions with respect to the most recently learned task.

| Targets "the brown dog is running on the grass" - "brown dog is running in a field" | |
| --- | --- |
| **After training task 1 (Scene):** | **After training task 4 (Instruments):** |
| **FT** a group of people are walking through a snowy mountain | **FT** a man in a black shirt and a man in a \<unk\> \<unk\> ... |
| **EWC** a group of people are walking through a snowy mountain | **EWC** a group of people are standing on a \<unk\> in the snow |
| **LwF** a group of people are walking through a snowy mountain | **LwF** a group of people are standing in front of a \<unk\> |
| **RATT** a group of people are walking down a dirt road | **RATT** a group of people are walking down a dirt road |
| Targets "a group of people walk through the desert" - "a group of 5 people are walking toward the mountains" | |
| **After training task 2 (Animals):** | **After training task 4 (Instruments):** |
| **FT** a brown dog is running through a field | **FT** a dog is playing a frisbee in the grass |
| **EWC** a brown dog is running through the grass | **EWC** a black dog is playing in the grass |
| **LwF** a man in a blue shirt is running in the grass | **LwF** a man playing with a dog |
| **RATT** a brown dog is running through a grassy field | **RATT** a brown dog runs through the grass |
| Targets "a race car speeding on the track" - "red and white car rounds a corner on a racetrack" | |
| **After training task 3 (Vehicles):** | **After training task 4 (Instruments):** |
| **FT** a race car is driving down a road | **FT** a \<unk\> \<unk\> \<unk\> \<unk\> \<unk\> \<unk\> \<unk\> ... |
| **EWC** a man in a red shirt is riding a bike | **EWC** a man in a red shirt is riding a motorcycle |
| **LwF** a man in a red shirt is riding a motorcycle on a street | **LwF** a man in a red shirt is playing a \<unk\> |
| **RATT** a car is racing on a track | **RATT** a car is racing on a track |

Figure 11: Captioning results Flickr30k. Images and target captions belong to a specific task and captions are generated by all techniques after training the correct task (left) and a later task (right).

| | |
| --- | --- |
| **Targets** | a cat on a leather chair next to remotes |
| | a cat sitting on a couch with two remote controls |
| | a cat sitting on top of a brown leather chair |
| | a cat is sitting on a leather couch next to two remotes |
| | a cat sitting in the chair with two remotes on the arm of the chair |
| **Predictions** | **After training task 5 (interior)** |
| | **FT:** a cat laying on top of a couch next to a remote |
| | **EWC:** a cat sitting on top of a laptop computer |
| | **LWF:** a cat sitting on a couch with a \<unk\> on it |
| | **RATT:** a cat is laying on a bed with a cat |

| | |
| --- | --- |
| **Targets** | a room filled with different types of items all around |
| | a stove top oven with multiple \<unk\> sitting in a kitchen |
| | this is a high tech stove that has many compartment and drawers |
| | a kitchen scene with focus on an old fashioned oven |
| | a silver oven a silver sink and a black stove |
| **Predictions** | **After training task 5 (interior)** |
| | **FT:** a kitchen with a stove and a microwave |
| | **EWC:** a white refrigerator is sitting on a bed |
| | **LWF:** a kitchen with a stove and oven of a microwave |
| | **RATT:** a white refrigerator freezer sitting on top of a stove |

Figure 12: Examples of images from MS-COCO dataset for which fine tuning achieve better results than the proposed method. These images are taken from the last task, so there is no catastrophic interference.

## Footnotes

*Code for experiments and task splits available here: https://github.com/delchiaro/RATT