[Reviews · NeurIPS 2020]

Review 1

Summary and Contributions: This paper takes a systematic look at continual learning of LSTM-based models for image captioning. It adapts continual learning approaches based on weight regularization and knowledge distillation to image captioning. Besides, this paper proposes an attention-based approach, called Recurrent Attention to Transient Tasks (RATT). This approach is evaluated in incremental image captioning on two new continual learning benchmarks based on MS-COCO and Flickr30, showing no forgetting of previously learned tasks.

Strengths: 1. This paper is the first work to focus on the continual learning of recurrent models applied to image captioning. 2. The proposed approach named RATT is able to sequentially learn five captioning tasks while incurring no forgetting of previously learned ones, which is supported by experiments on two new continual learning benchmarks defined using the MS-COCO and Flickr30 datasets. 3. The experiments result in Table2 and Table3 show that the method is effective, the forgetting rate has dropped significantly.

Weaknesses: 1. overall, the paper lacks enough innovation, the method just modified HAT[1] to recurrent networks for image captioning 2. There are many excellent image captioning models. But the baseline compared with the method is too simple, the experiments results lack sufficient conviction. [1] Joan Serra, Didac Suris, Marius Miron, and Alexandros Karatzoglou. Overcoming catastrophic forgetting with hard attention to the task. In International Conference on Machine Learning (ICML), 2018.

Correctness: Yes, The author provides detailed comparative experiments.

Clarity: This paper is mainly well written, although it contains many trivial formulas, like (6) - (11), which is the formulas of LSTM. And the structure may not be reasonable, as the related work, which is closely related to chapter 2, should be written before chapter 2.

Relation to Prior Work: Almost yes. The related work of catastrophic forgetting is well written, but the related works of image captioning is very few.

Reproducibility: Yes

Additional Feedback:


Review 2

Summary and Contributions: The paper studies the continual learning of recurrent networks on the image captioning task. It proposed the novel Recurrent Attention to Transient Tasks (RATT) method inspired by previous attention-based continual learning approaches. Experiments on the COCO dataset and the Flickr30K dataset show impressive results, where RATT achieves almost zero (or even negative) forgetting.

Strengths: The paper is one of the first to study continual learning in recurrent settings and shows promising performance on the image captioning task. It proposes RATT, a novel approach for recurrent continual learning based on attentional masking, inspired by the previous HAT method. In its proposed method, three masks (a_x, a_h, and a_s) to embedding, hidden state, and vocabulary are introduced, and in its ablation study, the paper shows that all these three components are helpful to the final continual learning performance. In addition to the proposed novel approach, the paper also explores adapting weight regularization and knowledge distillation-based approaches to the recurrent continual learning problem. In its experiments, the paper shows strong results, largely outperforming simple baselines (such as fine-tuning) and previous regularization or distillation-based approaches (EWC and LwF). It achieves almost zero forgetting on COCO and negative forgetting on Flicker30K, which is very impressive. Update: After reading the authors' response, I am convinced by the authors' claims and would recommend accepting the paper. Regarding the novelty of this paper, first, since there is no previous work on continual learning in the image captioning domain, explorations in this direction should constitute a novelty by itself. Second, while the proposed methodology is inspired by HAT, it involves non-trivial adaptation of HAT to the image captioning task, such as introducing three different masks (a_x, a_h, and a_s) to embedding, hidden state, and vocabulary. The paper also provided detailed analyses on these aspects. Hence, I believe the paper is novel in both the problem it addresses and its methodology.

Weaknesses: I do not think the paper has any major weaknesses. However, it is widely known that automatic metrics like BLEU and CIDEr often do not align well with human judgment on the quality of the captions. It would be better if the paper can introduce human evaluation of the generated samples. Besides, the captioning model used in this work is a very simple model compared to the state-of-the-art on image captioning. It would be better if the paper could show that the proposed RATT method generalizes well to more recent captioning methods such as AoA [A]. [A] Huang, Lun, et al. "Attention on attention for image captioning." Proceedings of the IEEE International Conference on Computer Vision. 2019.

Correctness: The paper's claims and its method are correct.

Clarity: The paper is well-written.

Relation to Prior Work: Yes, the paper clearly discusses (and distinguishes itself from) previous work on continual learning.

Reproducibility: Yes

Additional Feedback: After reading the authors' response, I am convinced by the authors' claims and would recommend accepting the paper. Regarding the novelty of this paper, first, since there is no previous work on continual learning in the image captioning domain, explorations in this direction should constitute a novelty by itself. Second, while the proposed methodology is inspired by HAT, it involves non-trivial adaptation of HAT to the image captioning task, such as introducing three different masks (a_x, a_h, and a_s) to embedding, hidden state, and vocabulary. The paper also provided detailed analyses on these aspects. Hence, I believe the paper is novel in both the problem it addresses and its methodology.


Review 3

Summary and Contributions: This paper presents a new method RATT on solving continual learning problem for LSTM-based models on image captioning, which self-defines two continual learning benchmarks and achieves promising results on it.

Strengths: 1.This paper first extends traditional continual learning problem on image captioning and creates two new task splits based on COCO and Flicker30K for with thorough evaluation. 2. The empirical evaluation shows RATT is effective in preventing catastrophic forgetting compared to other recurrent continual learning methods, which even obtains zero or nearly zero forgetting. 3. The paper has a clear structure and is well organized.

Weaknesses: 1. For the task split on COCO, can the authors provide the overlapping words percent between different tasks? It is meaningful to know the disjoint degree between them. 2. In Table 2, the proposes RATT achieves nearly zero forgetting. As each task share common words, can the authors more clearly explain why the new training task does not have any influences on the old task (the trainable weights for common parts between different tasks are fixed?). 3. What are the limitation cases or failure cases of the proposed method. 4. The baseline (show and tell) is proposed about five years ago, whose performance is far behind current SOTA models using LSTM, which needs a mention or comparison, such as [a,b,c]. And more and more methods using Transformer [c, d] instead of LSTM for captioning. The related work for image captioning should be more complete and up-to-date. [a] Bottom-up and top-down attention for image captioning and visual question answering. CVPR,2018. [b] "Regularizing rnns for caption generation by reconstructing the past with the present." CVPR. 2018. [c] Reflective Decoding Network for Image Captioning. ICCV, 2019. [d] Meshed-Memory Transformer for Image Captioning. CVPR, 2020. [e] Image captioning: Transforming objects into words." Advances in Neural Information Processing Systems. 2019.

Correctness: The methods using attention mask is reasonable and empirical experiment analysis is adequate.

Clarity: Yes. This paper is well organized and well written.

Relation to Prior Work: Yes, the Catastrophic forgetting part in the related work section has a clear discussion.

Reproducibility: Yes

Additional Feedback: 1.Please consider a more clear analysis/explanation on zero forgetting and why LwF performs much better on Flicker30K than COCO. 2.Please consider a mention or discussion with the SOTA captions methods in recent years to complete the related work section. It will be better if the author can incorporate the performance of their method using more recent captioning models which also use LSTM. 3. The max score for Interior on CIDEr in Table 2 belongs to FT and Line280 should replace word "increase" to "decrease"?

[Author Response · NeurIPS 2020]

We thank the reviewers for their constructive feedback and will incorporate all input in the final version. All three
appreciate the effectiveness of our approach, as well as novelty of our work being the first to take a systematic look at
continual learning of LSTM-based captioning networks. We respond below to specific points raised by the reviewers.

**On novelty (R2).** Ours is the first work considering Continual Learning (CL) of image captioning models, and we are
among the first to consider CL for recurrent networks. Extending attention masking to recurrent networks for transient
tasks required careful design and non-trivial modifications. We propose two task-dependent masks: one on the hidden
state and one shared mask on the word and visual embedding. We also use a fixed mask on the classifier which, since
we consider transient tasks, can overlap between tasks. Finally, the backward masking on word embedding described in
lines 169-173 is a key difference from HAT that allows RATT to leave more weights free for future tasks where vanilla
HAT freeze parts of the word embedding even for non-active words.

These are fundamental differences between our approach and HAT. Our results show that in more difficult transient
setting excellent results can still be obtained if task-aware masking is carefully applied in different ways to different
parts of the network. Our ablation study in Figure 2 shows the importance of all masks to obtaining the best results. We
hope that RATT will lead to similar techniques for continual learning in other domains with recurrent networks (e.g.
machine translation, multi-label estimation, and visual question answering). In addition, we propose two new setups for
continual image captioning, which together with code for all methods will be made public upon acceptance.

**On vocabulary overlap (R4).** As suggested
by R4, we computed word overlaps between
tasks for our MS-COCO splits (shown in the
table to the right). From this breakdown we
see that the task vocabularies are approxi-

|   | T | A | S | F | I |
|---|---|---|---|---|---|
| **T** | 3,116 (100.0%) | 1,499 (48.11%) | 1,400 (44.93%) | 1,222 (39.22%) | 1,957 (62.80%) |
| **A** | 1,499 (48.11%) | 2,178 (100.0%) | 1,175 (53.95%) | 1,025 (47.06%) | 1,492 (68.50%) |
| **S** | 1,400 (44.93%) | 1,175 (53.95%) | 1,967 (100.0%) | 933 (47.43%) | 1,355 (68.89%) |
| **F** | 1,222 (39.22%) | 1,025 (47.06%) | 933 (47.43%) | 2,235 (100.0%) | 1,530 (68.46%) |
| **I** | 1,957 (62.80%) | 1,492 (68.50%) | 1,355 (68.89%) | 1,530 (68.46%) | 3,741 (100.0%) |

mately the same size (between around 2,000 and 3,000 words), and there is significant overlap between all tasks.

**On zero forgetting in RATT (R4).** Some words are shared between tasks in the last layer, however in the second-to-last
layer tasks have *different* attention masks. Thus, weights connecting neurons in the task mask in the second-to-last layer
to those in the last are used when computing the probability of a word. These different connection pathways ensure
that words can be adapted to their usage in new tasks without causing forgetting in previous ones – even when task
vocabulary overlap is significant. During evaluation the task embedding forces use of only weights that were not used
by future tasks. We will add elements of this analysis in the final version.

**On LwF performance (R4).** LwF is known to perform poorly for image classification when there are large domain
shifts between tasks (e.g. from flowers to airplanes) and better when domains are more closely related [Aljundi, CVPR
2017]. This could be why LwF performs better on Flicker30K, which uses *incremental visual categories* (and not
*visually-disjoint task splits* like MS-COCO). In Flickr30k new images with already seen visual concepts from previously
learned tasks can occur. Such images facilitate knowledge transfer with LwF, leading to improved performance.

**On captioning quality metrics (R3).** We performed an evaluation
based on human quality judgments using 200 images (40 from each
task) from the MS-COCO test splits. We generated captions with

|   | T | A | S | F | I |
|---|---|---|---|---|---|
| **RATT vs EWC** | 75.0% | 77.5% | 72.5% | 85.0% | 57.5% |
| **RATT vs LwF** | 77.5% | 82.5% | 75.0% | 62.5% | 47.5% |

RATT, EWC, and LwF after training on the last task and then presented ten users with an image and RATT and baseline
caption in random order. Users were asked (using forced choice) to select which caption best represents image content.
The percentage of users who chose RATT over the baseline are given in the table to the right. These results confirm that
RATT is superior on all tasks. We will expand this to include Flickr30k and more human judgments in the final version.

**On limitations and failure modes (R4).** A drawback of RATT is that at some point network capacity is exhausted and
there are no un-attended neurons left, meaning that the network can no longer adapt to new tasks. In this case either
network growing techniques should be considered or attention to previous tasks could be relaxed which will lead to
forgetting. We did not observe this yet in the experiments we performed, but will include a discussion in the article.

**On our choice of captioning model (R2, R3, R4).** CL studies methods to mitigate catastrophic forgetting, and the vast
majority concentrate on relatively simple, feed-forward CNNs for image classification. They use simple architectures
(e.g. ResNet18, ResNet34) in order to focus on the effects of catastrophic forgetting in continual learning. Similarly, we
deliberately chose a simple architecture to focus on the adaptation of continual learning techniques to a new problem
using a recurrent architecture (LSTM). Our goal was not to achieve the state-of-the-art captioning performance, but
rather to systematically study a set of CL techniques and adapt them to an RNN. Modules like attention mechanisms
will surely also suffer from forgetting, and while this is interesting to study on its own, RATT is an important first step
toward understanding and mitigating the complex problem of catastrophic forgetting in recurrent captioning networks.

As suggested by R4, we will expand the discussion of more recent captioning models in the related work section
(which, as suggested by R2, we will move to the Section 2) of the final version of this work, and return to discuss in the
conclusions the ramifications of catastrophic forgetting in captioning models with features like attention mechanisms.

[Meta-Review · NeurIPS 2020]

The paper received two accept reviews and one borderline reject [R1]. The main concern of R1 is the paper relies on simple/not the most recent approaches for both captioning and continual learning. The other reviewers and I agree to that but believe that for one of the first papers in continual learning for captioning that this is reasonable, even if it is not optimal. R1 did not respond after the rebuttal. The reviewers appreciate the the paper's contributions, including 1) First paper in continual learning in image captioning. 2) The experiment evaluation both automatic and with human evaluation 3) Effective approach for attention masking (adapted from HAT to RNN). 4) The caption model, while not SOTA, is acceptable because of its simplicity and representativeness. I agree with this evaluation and accept, however, I expect the authors to include the clarifications and improvements suggested by the reviewers and made in the author response, including clearly describing the technical difference to HAT early on in the paper. I encourage the authors to include results for a more recent can powerful captioning model (e.g. based on region features instead of conv features) in the final version as also suggested by all reviewers. PS: The authors should consider including a discussion of recent/concurrent works in the space: https://arxiv.org/pdf/1909.08745.pdf https://nips2018vigil.github.io/static/papers/accepted/18.pdf https://arxiv.org/pdf/2001.01578.pdf https://arxiv.org/pdf/2005.00785.pdf